

# Rates of palaeoecological change can inform ecosystem restoration

Walter Finsinger[1], Christian Bigler[2], Christoph Schwörer[3], Willy Tinner[3]

[1]ISEM, University of Montpellier, CNRS, IRD, Montpellier, FR-34095, France
[2]Department of Ecology and Environmental Science, Umeå University, Umeå, SE-90187, Sweden
[3]Institute of Plant Sciences and Oeschger Center for Climate Change Research, University of Bern, Bern, CH-3013, Switzerland

*Correspondence to*: Walter Finsinger (walter.finsinger@umontpellier.fr)

**Abstract**

Accelerations of ecosystem transformation raise concerns, to the extent that high rates of ecological change may be regarded
amongst the most important ongoing imbalances in the Earth system. Here, we used high-resolution pollen and diatom assemblages and associated ecological indicators (the sum of tree and shrub pollen and diatom-inferred total phosphorus concentrations as proxies for tree cover and lake-water eutrophication, respectively) spanning the past 150 years to emphasise that rate-of-change records based on compositional data may document transformations having substantially different causes and outcomes. To characterize rates of change also in terms of other key ecosystem features, we quantified
for both ecological indicators (i) the percentage of change per-unit-time, (ii) the percentage of change relative to a baseline level, and (iii) the rate of percentage change per-unit-time relative to a baseline level, taking into account the irregular spacing of palaeoecological data. These measures document how quickly specific facets of nature changed, their trajectory, as well as their status in terms of palaeoecological indicators. Ultimately, some past accelerations of community transformation may document the potential of ecosystems to rapidly recover important ecological attributes and functions. In
this context, insights from palaeoecological records may be useful to accelerate ecosystem restoration.

## 1 Introduction

Changes in community composition have long captured the attention of palaeoecologists as they often unfold transformations reflecting biotic responses to drivers of ecosystem change (e.g. climatic changes, human impact, competition, and changing disturbance regimes) (Birks, 2019). Amongst the different ways to explore and quantify past
community transformations, rates of ecological change appear as particularly useful to estimate both how quickly and how much communities changed in the past, thereby allowing to detect accelerations and slowdowns of ecological transformation (Steffen et al., 2015). Though the idea of irregular rates of change in nature was already mentioned in the Principles of Geology (Lyell, 1854) – e.g. chapter XLII: "Rate of change of species cannot be uniform" and "From the wearing through of an isthmus" – rates of palaeoecological change were formally introduced to Quaternary palaeoecology only more than a
century later (Jacobson and Grimm, 1986) when chronologies of sedimentary records were better constrained.



Late Quaternary records of rates of palaeoecological change based on pollen sequences, and thus by inference vegetation compositional turnover, often show two main periods of acceleration (Jacobson and Grimm, 1986; Lotter et al., 1992; Bennett and Humphry, 1995; Seddon et al., 2015; Finsinger et al., 2017; Nogué et al., 2021): one period of acceleration centred on the last deglaciation (c. 14,600-10,000 cal BP) when rapid and high-amplitude climate-driven vegetation changes

occurred, and another acceleration during recent millennia under smaller-amplitude climate changes and increased human pressure. These patterns were recently confirmed using records around the globe (Mottl et al., 2021a) and applying new and improved approaches to estimate rates of palaeoecological change (Mottl et al., 2021b). Importantly, that study clearly documents unprecedented rates of ecological change in recent millennia that exceeded rates of climate-driven vegetation change during the last deglaciation, thereby demonstrating that humans undoubtedly can be viewed as a potent force capable

of driving large and rapid ecological transformations (Overpeck and Breshears, 2021) already before the onset of industrialization.

However, despite the concerns for the ongoing climate-driven and human-driven acceleration of ecological transformation (Steffen et al., 2015; Jouffray et al., 2020; Purvis et al., 2019), rates of change based on community composition (Jacobson and Grimm, 1986) unfold only one facet of ecosystem change (Purvis et al., 2019). Thus, rising rates of change based on

assemblage records may document community transformations having substantially different underlying causes and, most importantly, different outcomes (Fig. 1). Yet, for a sound assessment of trends in nature, for instance from the perspective of ecological restoration (Clewell and Aronson, 2013), it may also be important to explore how other ecosystem properties changed (Purvis et al., 2019).

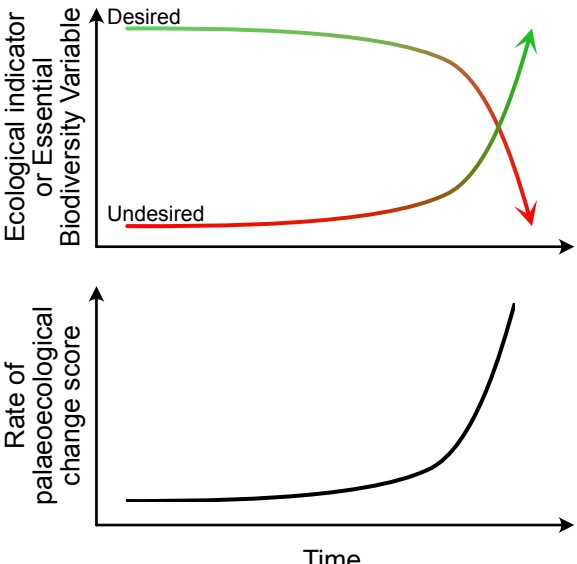

**Figure 1: Schematic model to illustrate the relationship between palaeoecological indicators characterizing the ecosystem state and its trajectory, and the resulting rate of palaeoecological change based on compositional data. The rate of palaeoecological change may rise irrespective of how and in which direction ecosystems changed, as it estimates the absolute amount of compositional change.**



Given these theoretical and empirical premises, the aim of the present contribution is to emphasise the importance of
characterizing both the direction and the rate of palaeoecological changes in terms of key features of ecosystems rather than
solely on community composition. To do this, we build on recent work of the Intergovernmental Science-Policy Platform on
Biodiversity and Ecosystem Services (IPBES, 2019) that used Essential Biodiversity Variables (EBVs; Pereira et al., 2013)
and ecological indicators as a basis for the assessment of both the trends in nature and the current status of nature (Purvis et
al., 2019). The examples chosen concern pollen and diatom records spanning the past 150 years, and therefore by inference
post-industrial transformations of both terrestrial and freshwater ecosystems. However, the problems discussed here may
relate to the interpretation of rates of palaeoecological change based on other proxy types and time intervals.

## 2 The context

We draw on pollen and diatom records from Lago Grande di Avigliana (Piedmont, southern European Alps, Italy) that
document vegetation and limnological changes occurring during the past c. 150 years (Finsinger et al., 2006). The records
have a well-established chronology that is based on annual varve counts supported by short-lived radionuclide measurements
back to c. 1930 CE and by biostratigraphic control points beyond that age (Finsinger et al., 2006; van der Knaap et al., 2000).
The inter-sample distance for the pollen and diatom records is overall <14 years (median inter-sample distances: 3 and 4
years, respectively; Fig. A1), and <5 years for the time interval 1940-2000 CE.

The pollen record documents an increase of tree cover starting around 1960 CE (Fig. 2a), as also observed in other high-
resolution pollen records from the European Alps that provide evidence for the spread of natural woodlands mainly as a
result of the abandonment of agriculture in marginal areas (Tinner et al., 1998; van der Knaap et al., 2000; Brugger et al.,
2021) and reforestation programs and afforestation actions of the EU in the past few decades (Fuchs et al., 2013; Brugger et
al., 2021). At Lago Grande di Avigliana this is supported by the decrease of cultivated species such as *Juglans* and *Secale*
and the enlargement of the wooded area by the spread of different taxa, such as pioneer taxa colonizing abandoned fields and
meadows (*Betula*), trees growing on wetter sites such as the lake shores (*Salix*), arboreal plants of the mixed oak forest
(*Fraxinus*, *Quercus*), ornamental trees (*Platanus*), as well as late-successional shade-tolerant trees (*Fagus*) (Tinner et al.,
1998).

In the freshwater lake ecosystem, diatom assemblages substantially changed at c. 1950-1960 CE and diatom-inferred total
phosphorus concentration (DI-TP) rapidly rose. DI-TP culminated in the late 1960s (Fig. 2b) as a result of the sewage
discharge from the bursting and unmanaged urban drain network. Subsequently, DI-TP values decreased indicating a rapid
recovery of the freshwater ecosystem due to conservation measures, such as the drain-network deflection of sewage
discharge that likely reduced phosphorus loads (Finsinger et al., 2006). The DI-TP record illustrates the eutrophication
history common to several low elevation lakes in the European Alps over the 19th-20th centuries (Lotter, 1998; Marchetto et
al., 2004; Bigler et al., 2007).





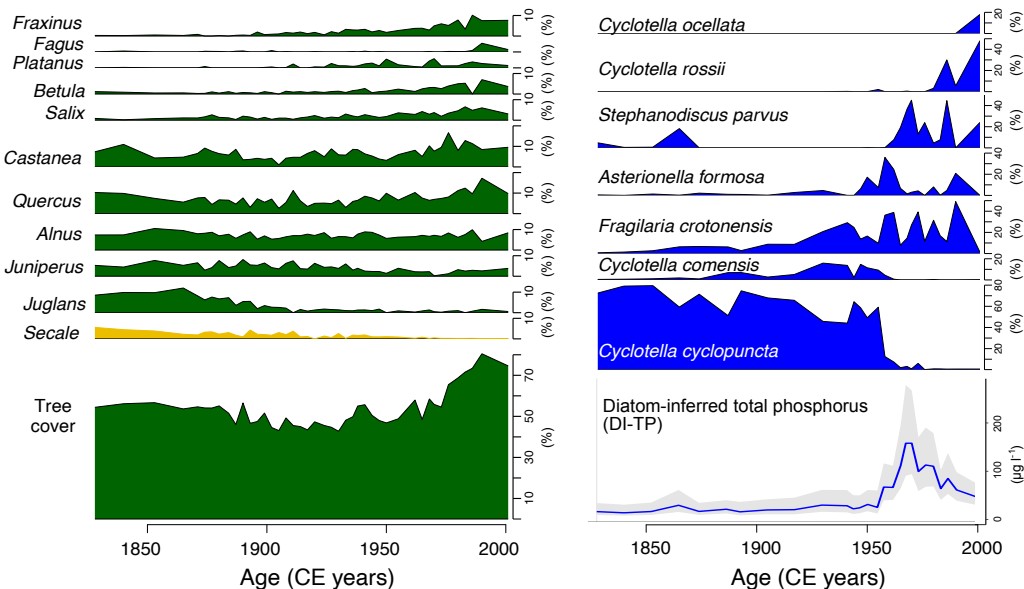

**Figure 2: Synthetic illustration of community composition changes for the terrestrial vegetation (left) and the aquatic diatom flora (right) as documented by pollen and diatom percentages from sediments of Lago Grande di Avigliana (Italy) for the past c. 150 years, as well as pollen-inferred tree cover (%) and diatom-inferred total phosphorous (TP, µg l⁻¹) concentration (from Finsinger et al., 2006).**

## 3 Methods

To account for the irregular spacing of palaeoecological records, rate of palaeoecological change (hereafter RoC) scores were calculated with the R-Ratepoll package (Mottl et al., 2021b) (see Appendix A for further details). In addition, we devised two custom-coded functions in the R environment (R Core Team, 2020) to unfold both how quickly and in which direction palaeoecological indicators changed. To explore the trends of palaeoecological indicators for the terrestrial and freshwater ecosystems, we used the relative abundance of tree and shrub pollen as a proxy for tree cover (Lang et al., 2023), and the diatom-inferred total phosphorous (DI-TP) concentrations (Finsinger et al., 2006) that illustrates the effects of cultural eutrophication and freshwater restoration (Lotter et al., 1997; Smol, 2008). Specifically, we quantified (i) the rate of change of the indicators as a percentage of change per-unit-time to show how quickly the system changed throughout and its direction, (ii) the percentage of change relative to the inferred or estimated baseline level to show how much was present after (and if applicable also before) the baseline period, and (iii) the rate of percentage change per-unit-time relative to the inferred or estimated baseline level (Purvis et al., 2019). See Appendix A for further details.



# 4 Results

## 4.1 Rates of palaeoecological change

The pollen and diatom assemblages changed substantially during the second half of the 21st century, and significant RoC
peaks were detected in both records between 1950 and 2000 CE (Fig. 3a-b). RoC scores increased faster for diatom
assemblages and the magnitude of change was about two times larger for the diatom than for the pollen assemblages. The
shorter generation times of diatoms and the fact that the lake is a closed ecosystem may be important, though not exclusive,
factors contributing to the different velocity and magnitude of change (Ammann et al., 2000). The RoC records also differ in
their trajectory after 1950 CE, as RoC scores of diatom assemblages were persistently high until 2000 CE, whereas RoC
scores of pollen assemblages decreased in the most recent decades.

## 4.2 Trends of palaeoecological indicators

The per-decade mean percentage change of tree cover (Fig. 3c) shows transient variations (±20% per decade) until 1950 CE
followed by a 30-year long period of persistently positive changes (up to 30% per decade) between 1950 and 1990 CE. DI-
TP varied even less before 1950 CE (Fig. 3d) when compared to its highest mean percentage change (up to 300 % per
decade) around 1960 CE. However, the per-decade mean percentage change of DI-TP persistently decreased after 1960 CE
(up to -50% per decade).

Compared to the average inferred baseline values of palaeoecological indicators, around 1990-2000 CE the tree cover was up
to 40% higher than during 1970-1979 CE (Fig. 3e) and up to 60% higher than during 1940-1950 CE (Fig. A2). Between
1970-1979 and 2000 CE, the per-decade rate of change in tree cover was 10% (Fig. 3e). DI-TP values were up to 1000%
higher than during the baseline period 1800-1899 CE and the per-decade rate of change of DI-TP was 17% between 1800-
1899 and 2000 CE (Fig. 3f).





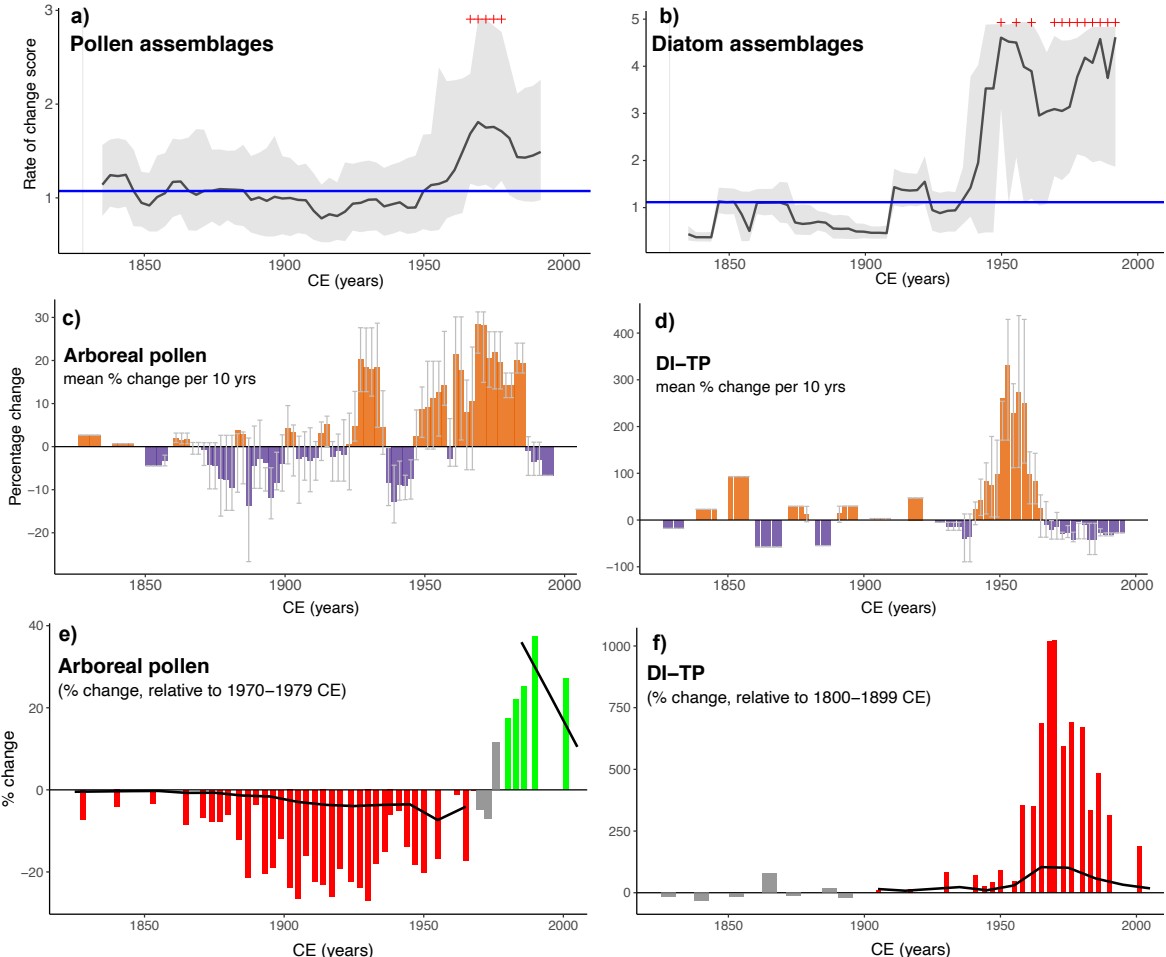

**Figure 3: Comparison between rates of palaeoecological change based on (a) the pollen and (b) the diatom assemblages from sediments of Lago Grande di Avigliana, and (c-f) the trends of palaeoecological indicators for the terrestrial and aquatic**
**ecosystems (arboreal pollen percentages indicative of tree cover and diatom-inferred total phosphorous concentration (DI-TP), respectively) for the past c. 150 years. The panels in the middle (c-d) document how quickly and in which direction the palaeoecological indicators changed as the average decadal rate of percentage change (coloured bars: mean values; whiskers: first and third quartiles). The bottom panels (e-f) show how much of the indicator was present after (and if applicable also before) the baseline period (grey bars), as well as how quickly and in which direction the palaeoecological indicators shifted from the average**
**baseline value (black continuous line). In panels e) and f), the bars are colour-coded based on whether positive changes relative to the average baseline value are desired (green) or undesired (red).**

## 5 Discussion and conclusions

Assessments of past rates of ecological change are important to explore how quickly ecosystems changed through time and highlight accelerations of ecosystem transformation (Overpeck and Breshears, 2021). Currently, high expected rates of
ecological change in response to predicted high rates of climate change raise concerns, as species' and ecosystems' capacity to respond adaptively may be outpaced (Williams et al., 2021). Moreover, human actions appear as significant drivers for



rising rates of change of vegetation composition (Mottl et al., 2021a), to the extent that high rates of ecological changes may be regarded amongst the most important ongoing imbalances in the Earth system (Albert et al., 2023).

We found that RoC scores of pollen assemblages significantly peaked at the time of the spread of natural woodlands as a
result of land-use changes including abandonment of agricultural lands in marginal areas (Fig. 2 and 3a), indicating that high RoC scores can also document shifts towards less degraded conditions. Likewise, diatom assemblages significantly changed both when lake-water eutrophication increased as well as when the freshwater system recovered towards more desired total phosphorus concentrations (Fig. 2 and 3b). Thus, rises and significant peaks of rate-of-palaeoecological-change (RoC) records may occur irrespective of the underlying trajectory of ecosystems (Fig. 1), thereby supporting the view that RoC
records based on compositional data are ambiguous and should be interpreted in the context of the nature of changes.

By contrast, the assessment of the rate of change of ecological indicators allows assessing how quickly and in which direction features of ecosystems changed through time. For instance, the per-decade percentage change of the arboreal pollen documents a net gain in tree cover when vegetation composition significantly changed (Fig. 3c). Likewise, the mean percentage change of the DI-TP record documents the rate of change as well as the changing trajectory of lake eutrophication
during the time interval of significant diatom-assemblage changes (Fig. 3d). Thus, these measures of ecosystem trajectories based on palaeoecological indicators document how the 19th-20th century's and ongoing land-use changes in the lowlands of the European Alps reversed longer-term trends of ecosystem degradation (Birks and Tinner, 2016), both accidentally (e.g. land abandonment) and intentionally (lake restoration, forest protection and/or restoration).

Such assessments are consistent with the way the Intergovernmental Science-Policy Platform on Biodiversity and Ecosystem
Services (IPBES, 2019) assesses trends in nature on the basis of ecological indicators and Essential Biodiversity Variables (Pereira et al., 2013; Purvis et al., 2019), some of which can be estimated based on palaeoecological records and could complement RoC records. Admittedly, only one facet of nature was explored here for the terrestrial and the freshwater environments, respectively. Given that nature is too complex for its trends and status to be captured by one or a few indicators (Purvis et al., 2019), palaeoecological data may be explored to yield other indicators, such as the abundance of
invasive and exotic species, biodiversity measures, browsing and grazing intensity, fire activity, or environmental disturbance such as drought or frost (Tinner, 2023), to name a few. It may further be speculated that the ambiguity of RoC records is not limited to records for the 20th-21st centuries when restoration actions often reversed longer-term trends of ecosystem degradation. Previous changes in ecological community composition were also sometimes embedded in a longer-term framework of varying human footprints. For instance, pollen records from Europe often document for the late Holocene
alternating land-abandonment phases and more intensive land-use phases (Schibler et al., 1997; Tinner et al., 2003; Finsinger and Tinner, 2006; Rey et al., 2019). The transition from one of these phases to another often involved a more or less marked vegetation-community change, including the rise and fall of tree cover. In such cases, focussing solely on rate of compositional change may give an erratic picture of ecological changes.

Assessing the long-term development of palaeoecological indicators may also be useful to explore key features of
ecosystems relative to ecological baselines. Baselines are an integral part of the conceptual framework of the UN's decade



on ecosystem restoration (2021-2030) and are an asset of palaeoecological records (Willis et al., 2010; Nogué et al., 2022; Burge et al., 2023) as the current status of nature is often assessed with reference to relatively recent baselines (e.g. mostly after 1970 CE; Purvis et al., 2019). In this context, the pollen record indicates that tree cover was 60% higher around 1990-2000 CE than in 1940-1950 CE. Though the increase matches with a net gain in tree cover documented based on land-cover

datasets (25%) for Europe from 1950 to 2010 CE (Fuchs et al., 2013), it suggests the occurrence of a high spatial variability that could be further explored with a network of pollen records. Likewise, tree cover rose at an average per-decade rate of 10% between 1970-1979 and 2000 CE, suggesting that tree cover in the southern Alps rose about five times faster than the global average (2.1%; Purvis et al., 2019). Moreover, several diatom taxa that dominated the lakes' flora prior to the cultural eutrophication did not reappear during the recovery (Fig. 2), a feature often observed in palaeolimnological diatom records

(Smol, 2008). In contrast to methods based on community composition (e.g. Burge et al., 2023) that would very likely highlight the dissimilarity between the assemblages, information provided by the change of palaeoecological indicators gives "permission to accept transient environmental and ecological change" (Jackson and Hobbs, 2009).

The analysis of rates of compositional change together with those of palaeoecological indicators that document key features of nature (e.g. tree cover, freshwater eutrophication, diversity measures, and disturbance dynamics) thus may provide

insights into to the speed of ecosystem recovery as well as to conditions under which ecosystem recovery may and may not occur, thereby informing managers about the level of intervention needed in ecosystem restoration (Willis et al., 2010; Whitlock et al., 2018). Ultimately, past accelerations of community transformation may document the potential of ecosystems to shift toward more desired states and recover important ecological attributes such as their functions and diversity. Such insights may be useful to accelerate ecosystem restoration (Manzano et al., 2020; Edrisi and Abhilash, 2021;

Gillson, 2022).

**Appendix A:**

**A1 Material**

The pollen data from Lago Grande di Avigliana with associated metadata and chronology (Finsinger et al., 2019) were obtained from the Neotoma database (Giesecke et al., 2014; Williams et al., 2018) using the NeotomaExplorer App. The

pollen and diatom datasets were treated with custom-coded functions in the R environment (R Core Team, 2020) to adapt their format for the R-Ratepoll v1.2.1 package (Mottl et al., 2021b).

**A2 Methods**

**A2.1 Data pre-treatment and analysis**

The treatments for the pollen data include filtering the data to select only terrestrial pollen types (thus, excluding pollen and

spores from obligate aquatic plants), removing the *Humulus/Cannabis*-type and a sample having a low pollen count sum





(<195), and harmonizing the taxonomy following Mottl et al. (2021a). The diatom data was filtered to include only diatom taxa (thus, excluding Chrysophyte cysts). Initial exploratory data analysis also focussed on the assessment of the inter-sample distances (in years) of the samples (Fig. A1).

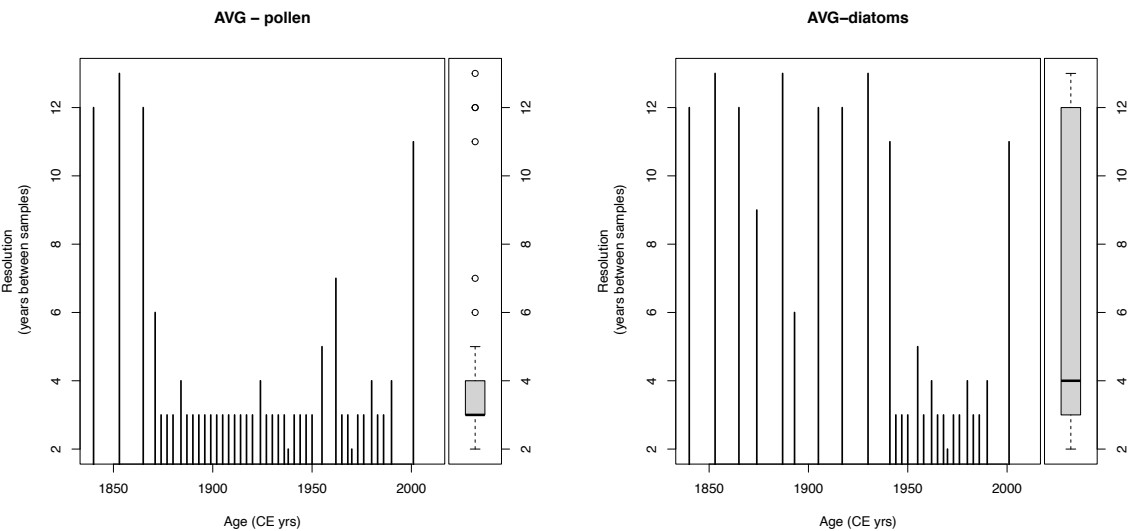

**Figure A1: Inter-sample distance for the pollen (left) and the diatom records (right) from the sediments of Lago Grande di Avigliana (Finsinger et al., 2006).**

**A2.2 Rates of palaeoecological change**

Rate of palaeoecological change (hereafter RoC) scores were calculated with the R-Ratepoll package (Mottl et al., 2021b) using the pollen and diatom counts with the following settings: age-weighted average smoothing, binning with 14-year long moving windows (thus, with larger working units than the largest inter-sample distance), shifting moving windows three times, random selection of samples from bins, random sampling without replacement to draw 195 individuals from each working unit, the chi-squared coefficient as the dissimilarity coefficient, transform the counts to proportions, and rescaling RoC score values to the RoC per 50 years. These calculations were reiterated 999 times, and thereafter significant RoC peaks were identified by comparing the 95[th] quantile of the RoC scores from all calculations and the median of all RoC scores from the whole sequence.

**A2.3 Trends of palaeoecological indicators**

Firstly, we quantified the rate of change of the palaeoecological indicator as a percentage of change per-unit-time, thereby showing how quickly the indicator changed and its trend (Purvis et al., 2019). Following Mottl et al. (2021b), we took into account the irregular spacing of palaeoecological records by binning with moving windows, random selection of samples from bins, and shifting moving windows (here five times). The random selection procedure was repeated 99 times for each



set of moving windows prior to calculate summary statistic values (i.e. the mean, the median, and the first and third quartiles). As we focussed on the higher sampling-resolution time interval, we used 10 years long moving windows. Secondly, we quantified the change in the indicator as a percentage of change relative to the inferred or estimated baseline level, thereby showing how much was present after the baseline period (Purvis et al., 2019) and, if applicable also how much

was present before the baseline period. Thirdly, we quantified the rate of percentage change per unit time (here a decade) relative to the inferred or estimated baseline level.

As baseline levels for the pollen-inferred tree cover, we selected the time interval 1970-1979 CE (following Purvis et al., 2023) as well as the time interval 1940-1950 CE (Fig. A2) to compare the results with documentary data (Fuchs et al., 2013). Likewise, we use temporally based reference conditions using paleo-reconstructions for the freshwater ecosystem, as

acknowledged in the European Water Framework Directive (WFD 2000/60/EC; Bennion et al., 2011; Smol, 2008). Specifically, for the DI-TP record we selected the reference time interval 1800-1899 CE, which pre-dates the earliest major step in lake-water eutrophication during the 19th-20th centuries in lowland lakes in the region of the European Alps (Bigler et al., 2007; Finsinger et al., 2006; Lotter, 1998; Marchetto et al., 2004).

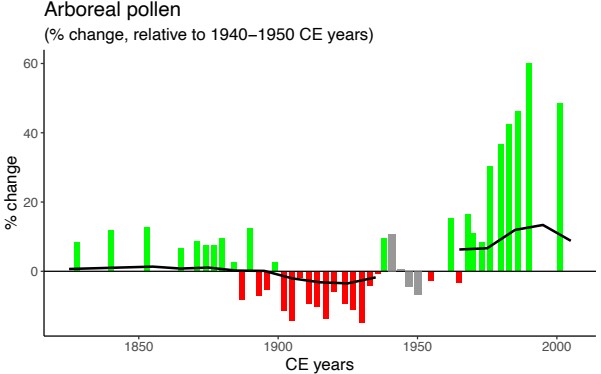

**Figure A2: Percentage change (green and red bars) and rate of percentage change per decade (black line) relative to 1940-1950 CE of arboreal pollen (grey bars), and thus by inference of tree cover.**

**Code and data availability**

The pollen data were obtained from the Neotoma database (Giesecke et al., 2014; Williams et al., 2018). The diatom record as well as all computer codes will be publicly available in suitable repositories (pangaea.de, zenodo.org), and are currently

available at https://doi.org/10.5281/zenodo.10075147.

**Author contributions**

WF conceived of the paper and was responsible for developing the code, generating all figures, and writing the original draft. The manuscript was reviewed and edited with contributions from all co-authors.



**Competing interests**

The authors declare that they have no conflict of interest.

**Acknowledgements**

Data were obtained from the Neotoma Paleoecology Database (http://www.neotomadb.org). The work of data contributors, data stewards, as well as the Neotoma, the European Pollen Database (EPD), and the Alpine Pollen Database (ALPADABA) communities is gratefully acknowledged.

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
