# Peer review of "Rates of palaeoecological change can inform ecosystem restoration"

_EGUsphere, 2023_

## Author Comment (AC1)

RC1: ['Comment on egusphere-2023-2623'](), Anonymous Referee #1, 22 Dec 2023
Citation: https://doi.org/10.5194/egusphere-2023-2623-RC1

The manuscript "Rates of palaeoecological change can inform ecosystem restoration" focuses on data analysis for the past 150 years from Lago Grande di Avigliana lake. This is a very well written manuscript dealing with relevant environmental concerns, such as biodiversity and its restoration. I think that one of the relevant messages that makes this paper very interesting is the documentation of past acceleration of community transformation and the use of palaeoecological indicators. I would also like to highlight that the authors made an effort to include policy context. Overall, the paper is clear, well-structured, methodologically robust, and with an excellent quality.

> Many thanks for the overall positive assessment of our manuscript.

However, there are some minor aspects that to me requires a bit of improvement.

1) why the authors focused on Lago Grande di Avigliana requires a bit more of explanation. I'm assuming that might be for the quality of the chronology and available data (some information on the section called "context"). However, to me, the manuscript could improve if the authors include a paragraph justifying the choice of study site and also why the results from this lake have the potential to help improving our understanding on restoration of ecosystems.

> Right, the manuscript was a little weak on the justification of the data set.

To address your comment, we have drafted a possible revision that may involve merging the '1 Introduction' with the '2 The context' sections and adding a sentence that sets the transition between the two sections (see underlined text here below):

"[…]. The examples chosen concern post-industrial transformations of both terrestrial and freshwater ecosystems, as documented by pollen and diatom records from a lake located in the forelands of the western Italian Alps (Finsinger et al., 2006). We chose these palaeoecological records because they have a high temporal resolution and a well-established chronology based on annual varve counts supported by short-lived radionuclide measurements and by biostratigraphic control points. Accordingly, these datasets (Fig. A1) disclose rapid environmental changes in terrestrial and aquatic ecosystems revealing compositional changes that occurred in conjunction with both undesired and desired changes of ecological properties during the past 150 years. Specifically, the pollen document an increase of tree cover […]"

2) I found methods a bit short and limited in detail. How about including some of the information currently placed in the appendix in the methods section? I would suggest at least to include in the main text A1 Material.

> Many thanks for the suggestion. We will move the text "A1 Material" in the Methods section of the main text.

Very interesting manuscript!

> Many thanks for the encouraging remark on the manuscript. Much obliged.

---

## Author Comment (AC2)

RC2: 'Comment on egusphere-2023-2623', Alistair W.R. Seddon, 08 Jan 2024
Citation: https://doi.org/10.5194/egusphere-2023-2623-RC2

The paper explores ideas related rate-of-change (RoC) analysis, using a published diatom and pollen dataset from the Italian Alps to move beyond RoC analysis of palaeoecological assemblages and towards ecological properties. They extend ideas of RoC analysis to look at the amount of change relative to baseline conditions. The main point is that whilst RoC analyses on assemblages can detect points of assemblage shifts, they can mask other important details (e.g. whether the changes are positive or negative in the context of management/conservation). In moving towards temporal analysis of ecological properties the authors are also making a small step towards acknowledging palaeoecology's potential contribution to assessing Essential Biodiversity Variables (EBVs).
I think the paper makes an important contribution since RoC curves (e.g. Mottl et al 2021) should always be assessed in the context of the ecological changes that are underlying them and this paper helps emphasise this point. The move towards highlighting the relevance of palaeoecology for IPBES and EBVs is also very welcome.
> Many thanks for the in-depth and positive appraisal of our manuscript.

One important caveat is that even some derived ecosystem properties (e.g. in this instance, tree cover inferred from arboreal pollen percentages) are also context dependent. Whilst increases in tree cover (i.e. 'positive' rates of change) might be viewed favourably in the context of the ecological restoration for this alpine ecosystem, in other locations (e.g. grasslands) increases in tree cover might not necessarily be representative of a positive change in terms of restoration (e.g. Veldmann et al. 2019, DOI: 10.1126/science.aay7976). Thus, some derived ecosystem properties are likely to be as context dependent as the pollen/diatom assemblages themselves. This is an important point to make if the approaches used on ecosystem properties are further generalized and I think this should be emphasized in the discussion.
> Thank you. Indeed, this is an important point that merits being mentioned. The caveat you highlight relates to the way changes of ecological properties are valued in the context of restoration. It seems therefore to be related to the long-term context relative to reference (or baseline) conditions. Thus, the caveat could build a transition between the paragraph delving on the limitations of our study due to the use of a limited set of ecological indicators, and the paragraph that addresses the baseline reference conditions.
We have taken your formulation as a basis to draft two sentences in the Discussion section. The sentences may read as follows:

"Moreover, it should be noted that ecosystem properties are also context dependent. Whilst increases in tree cover might be viewed favourably in the context of the ecological restoration for this ecosystem, in other locations (e.g. grasslands) increases in tree cover might not necessarily be representative of a positive change in terms of restoration (Veldman et al., 2019)."

A second critique, particularly from the pollen data, is that there are obvious potential representation / preservation issues that are not taken into account when RoCs on pollen assemblages are analysed, which can be addressed by correction by pollen production values and through methods such as REVEALS modelling. If palaeoecology is going to move towards EBVs then I think here would be as good a place as any to at least reference this critique in reference to recently published RoC analysis and the analysis of the ecological properties derived in this specific study.

> Yes, this is indeed an important point, particularly for the pollen data. We agree, the revised manuscript should acknowledge that the tree cover estimates based on pollen percentages, e.g. the net gain of 50-60% between 1940-1950 and 2000 CE, may be higher as potential representation issues were not taken into account. We have therefore drafted a sentence that could fit in the discussion section:

> "While the higher net gain documented by the pollen does not take into account potential representation issues, which could be addressed by correction for the differences in pollen productivity and dispersal (Sugita, 1994; Seppä, 2013), it suggests the occurrence of a high spatial variability that could be further explored with a network of pollen records."

The methods are quite brief in the main text, but even if I understand the papers is limited on space, I think you should move the paragraph about which baseline periods were selected from the appendix to the main methods section.

> Right, many thanks for the comment, which meets a suggestion made by reviewer #1 who also highlighted the excessive brevity of the main Methods section. We shall move the paragraph about which baseline periods were selected from the Appendix to the main Methods section.

It was also not clear to me how the black lines are calculated in Figures 3e-f,

> Many thanks for highlighting this weak point of the manuscript. The black line in Figures 3c-f was supposed to show the rate of change per unit time for the ecological indicator. Indeed, it appears we did not explain the calculation in sufficient depth. Moreover, we noted that some errors moseyed into the R code that we used to perform the analyses.

We thus shall add an explanation in section "A1.3 Trends of palaeoecological indicators" to clarify how the black line was calculated, and in addition made two corrections:

Firstly, we removed the black line for the period preceding the baseline interval (Figure 3e), as a rate of change that is calculated "backwards in time" may, indeed, be very awkward.

Secondly, we revised the formula to calculate the rate of change relative to the reference period. It now follows better the method used by Purvis et al. (2019) [see Supplementary material to Purvis et al. (2019) in SUP/GA/2.2 Chapter 2.2 Supplementary material (Nature) at https://www.ipbes.net/global-assessment].

Thus, the revised code first calculates the change between the mean reference value and the post-reference value, and thereafter divides that by the number of 'bins' between their dates to provide a per-unit-time rate of change. For instance, with bins equal to 10 years, the black line illustrates the per-decade rate of change.

As also stated by Purvis et al. (2019), it should be noted that this is just the average rate of net change over the time span being considered, whether or not the change was linear.

Please note that the code returns a plot with the black line and the grey vertical bars only in the case the user defined the reference period. Instead, if the user defined a reference value (or a

set of values), only the % change would be plotted (green and/or red vertical bars), as it would not be possible to calculate the time span separating the reference period from the target value. In addition, we revised the code to add the missing symbols in the legend of the Figure (the black line and the grey polygon).

See, for instance, the revised Figure 3e here below:

[Figure]

[and] The significance of using two different reference periods of the pollen data (e.g. Figures 3e and A2) is not really discussed in either the main text nor the appendix.
> Indeed, the choice of the reference period can strongly influence the outcomes of assessments on the status and trends. Specifically for the pollen data, we did choose two different reference periods (1970-1979 CE and 1940-1949 CE) as we aimed at comparing our results with documentary data (as was declared in the Appendix). However, it may be noteworthy to mention that the average per-decade rate of change in tree cover is about 9% both when setting the reference period at 1970-1979 (as of Purvis et al., 2019) as well as when setting the reference period at 1940-1950 (as of Fuchs et al., 2013). This evidence could support the notion of a net gain in tree cover documented based on land-cover datasets for Europe (Fuchs et al., 2013) for a longer time interval than the one considered in the IPBES assessment.

These revisions could help the reader.

**Cited literature**

Fuchs, R., Herold, M., Verburg, P. H., and Clevers, J. G. P. W.: A high-resolution and harmonized model approach for reconstructing and analysing historic land changes in Europe, Biogeosciences, 10, 1543–1559, https://doi.org/10.5194/bg-10-1543-2013, 2013.

Purvis, A., Molnár, Z., Obura, D., Ichii, K., Willis, K., Chettri, N., Dulloo, M., Hendry, A., Gabrielyan, B., Gutt, J., Jacob, U., Keskin, E., Niamir, A., Öztürk, B., Salimov, R., and Jaureguiberry, P.: Chapter 2.2 Status and Trends – Nature, in: Global assessment report of the Intergovernmental Science-Policy Platform on Biodiversity and Ecosystem Services, edited by: Brondizio, E. S., Settele, J., Díaz, S., and Ngo, H. T., IPBES secretariat, Bonn, Germany, 108, https://doi.org/10.5281/zenodo.3831674, 2019.

Seppä, H.: Pollen Analysis, Principles, in: Encyclopedia of Quaternary Science, Elsevier, 794–804, https://doi.org/10.1016/B978-0-444-53643-3.00171-0, 2013.

Sugita, S.: Pollen representation of vegetation in Quaternary sediments - theory and method in patchy vegetation, J. Ecol., 82, 881–897, 1994.

Veldman, J. W., Aleman, J. C., Alvarado, S. T., Anderson, T. M., Archibald, S., Bond, W. J., Boutton, T. W., Buchmann, N., Buisson, E., Canadell, J. G., Dechoum, M. D. S., Diaz-Toribio, M. H., Durigan, G., Ewel, J. J., Fernandes, G. W., Fidelis, A., Fleischman, F., Good, S. P., Griffith, D. M., Hermann, J.-M., Hoffmann, W. A., Le Stradic, S., Lehmann, C. E. R., Mahy, G., Nerlekar, A. N., Nippert, J. B., Noss, R. F., Osborne, C. P., Overbeck, G. E., Parr, C. L., Pausas, J. G., Pennington, R. T., Perring, M. P., Putz, F. E., Ratnam, J., Sankaran, M., Schmidt, I. B., Schmitt, C. B., Silveira, F. A. O., Staver, A. C., Stevens, N., Still, C. J., Strömberg, C. A. E., Temperton, V. M., Varner, J. M., and Zaloumis, N. P.: Comment on "The global tree restoration potential," Science, 366, eaay7976, https://doi.org/10.1126/science.aay7976, 2019.

---

## Author Response (AR1)

22 January 2024

Dear Prof. Paul Stoy (Co-editor-in-chief of *Biogeosciences*),
Dear Prof. Petr Kuneš (Associate editor of *Biogeosciences*),

Many thanks for being willing to reconsider the manuscript titled "*Rates of palaeoecological change can inform ecosystem restoration*" after major revisions.
Here, we submit to your and the reviewers' judgement our revised version of the manuscript.

To revise the manuscript, we have followed our replies to the referees comments (see AC1 and AC2 at https://egusphere.copernicus.org/preprints/2023/egusphere-2023-2623/#discussion).

In addition to the changes outlined in our replies, we:
- slightly shortened some of the sentences. As the reviewers suggested to move some parts from the Appendix to the main text, we tried keeping the length of the main text within the 2500-words limit. However, despite our efforts, the main text of the revised manuscript is slightly longer than what is recommended for BG Letters (~2600 words rather than 2500 words). We hope you may still consider it for final publication as BG Letters;
- replaced the terms "baseline level" and "baseline period" with the terms "reference level" and "reference period", respectively. As of the Oxford Dictionary of English, the term *baseline* refers to "*a minimum or starting point used for comparisons*". This term does not match well, as in the manuscript we do not necessarily refer to a minimum or starting point. We thought the terms *reference* [point], or *reference period*, which refers to "*a basis or standard for evaluation, assessment, or comparison; a criterion*", may be more suitable;
- revised one of the R functions we used to perform some of the analyses, in keeping with a suggestion made by a referee (specifically Prof. Alistair W.R. Seddon). Given the changes made to the computer codes, we have replaced Figure 3 as well as Figure A2. In addition, we have completed the documentation of the computer codes and the functions we devised for the analyses. An updated set of computer codes has been uploaded to GitHub and to the Zenodo repositories, where the data and codes are publicly available (https://zenodo.org/records/10075146).

Looking forward to the further evaluation of our manuscript.

Yours sincerely,

Walter Finsinger
(on behalf of all co-authors)

---

## Author Response (AR2)

30 January 2024

Dear Prof. Paul Stoy (Co-editor-in-chief of *Biogeosciences*),
Dear Prof. Petr Kuneš (Associate editor of *Biogeosciences*),

Many thanks for reconsidering the manuscript titled "*Rates of palaeoecological change can inform ecosystem restoration*" after the major revisions. Here, we submit our further revised version of the manuscript to your judgement.

Please, find here the point by point answer to the suggestions:

L. 129: It would be fair, according to policies, to also acknowledge the constituent databases of Neotoma, either the EPD or ALPADABA (or both if applicable). This also applies to l. 391.
> Yes, correct. Sorry for this oversight. We have corrected the sentence in the "Methods", the "Code and data availability", as well as in the "Acknowledgements" sections.
For the Methods section: *The pollen data from Lago Grande di Avigliana with associated metadata and chronology (Finsinger et al., 2019) were obtained from the Neotoma database (Williams et al., 2018) and its constituent database (European Pollen Database (EPD); Giesecke et al., 2014) using the NeotomaExplorer App.*
For the "Code and data availability" section: *The pollen data (Finsinger et al., 2019) were obtained from the Neotoma database (Giesecke et al., 2014; Williams et al., 2018) and its constituent database (European Pollen Database, EPD).*
For the "Acknowledgements": *Data were obtained from the Neotoma Paleoecology Database (http://www.neotomadb.org) and its constituent database (European Pollen Database, EPD). The work of data contributors, data stewards, as well as the Neotoma community is gratefully acknowledged.*

L. 130: "...datasets were treated with custom-coded functions in the R environment…" - is this really necessary here?
> Agreed. This is a detail that does not need to be mentioned in the main text, as it essentially refers to reshaping the data (reshape from a spreadsheet to a list; harmonising the pollen data; and filtering taxa). We have moved the sentence in the Appendix, where these data pre-treatments are better explained.

L. 131–132: This sentence could be merged with the previous to avoid text duplication and to shorten the text. Perhaps something like this: "The format of pollen and diatom datasets was adapted for the R-Ratepoll package (Mottl et al., 2021b), where the rate of palaeoecological change (hereafter RoC) scores were calculated in regular time bins."
> Many thanks for the suggestion to merge this sentence with the previous one. However, given the choice to move the previous sentence into the Appendix (see reply to previous comment), we would rather keep the text as it was formulated in the submitted manuscript (version R1).

Looking forward to the further evaluation of our manuscript.

Yours sincerely,

Walter Finsinger
(on behalf of all co-authors)

---

## Author Response (AR3)

09 February 2024

Dear Prof. Paul Stoy (Co-editor-in-chief of *Biogeosciences*),

Many thanks for considering the revised manuscript titled "*Rates of palaeoecological change can inform ecosystem restoration*" as publishable with corrections in BG Letters.
Here, we submit the production files.

**We have taken into account your comment (06 February 2024)**
*Please do not use red and green in figures at the same time and please replace all instances with a colourblind-friendly colour scheme.*
> Many thanks for pointing this out. We have modified Figure 1 and Figure 3 accordingly.
Specifically:
  - we selected colours from the diverging & Colour-vision-deficiency (CVD) friendly palette 'vik' (Crameri, F. (2018)), which has been used to illustrate annual global temperature change, amongst other (see https://s-ink.org/annual-global-temperature-change);
  - for panels 3e and 3f, we additionally added a legend to help all readers interpret the colours of the bar plots;
  - we have checked all the Figures with the Coblis – Color Blindness Simulator, as recommended in the BG submission guidelines;
  - finally, we have updated the R codes in the GitHub and Zenodo repositories.

Yours sincerely,

Walter Finsinger
(on behalf of all co-authors)